# Molecular Docking of Lac_CB10: Highlighting the Great Potential for Bioremediation of Recalcitrant Chemical Compounds by One Predicted Bacteroidetes CopA-Laccase

**DOI:** 10.3390/ijms24129785

**Published:** 2023-06-06

**Authors:** Bárbara Bonfá Buzzo, Silvana Giuliatti, Pâmela Aparecida Maldaner Pereira, Elisângela Soares Gomes-Pepe, Eliana Gertrudes de Macedo Lemos

**Affiliations:** 1Department of Agricultural and Environmental Biotechnology, Faculty of Agricultural and Veterinary Sciences (FCAV), São Paulo State University (UNESP), Jaboticabal 14884-900, SP, Brazil; barbara.bonfa@unesp.br (B.B.B.); elisangela.gomes@unesp.br (E.S.G.-P.); 2Graduate Program in Agricultural and Livestock Microbiology, UNESP, Jaboticabal 14884-900, SP, Brazil; 3Department of Genetics, Faculty of Medicine of Ribeirao Preto, Ribeirao Preto 13566-590, SP, Brazil; silvana@fmrp.usp.br; 4Molecular Biology Laboratory, Institute for Research in Bioenergy (IPBEN), Jaboticabal 14884-900, SP, Brazil; pamela.maldaner@unesp.br

**Keywords:** laccases, bioremediation of synthetic molecules, molecular docking simulation

## Abstract

Laccases are multicopper oxidases (MCOs) with a broad application spectrum, particularly in second-generation ethanol biotechnology and the bioremediation of xenobiotics and other highly recalcitrant compounds. Synthetic pesticides are xenobiotics with long environmental persistence, and the search for their effective bioremediation has mobilized the scientific community. Antibiotics, in turn, can pose severe risks for the emergence of multidrug-resistant microorganisms, as their frequent use for medical and veterinary purposes can generate constant selective pressure on the microbiota of urban and agricultural effluents. In the search for more efficient industrial processes, some bacterial laccases stand out for their tolerance to extreme physicochemical conditions and their fast generation cycles. Accordingly, to expand the range of effective approaches for the bioremediation of environmentally important compounds, the prospection of bacterial laccases was carried out from a custom genomic database. The best hit found in the genome of *Chitinophaga* sp. CB10, a Bacteroidetes isolate obtained from a biomass-degrading bacterial consortium, was subjected to in silico prediction, molecular docking, and molecular dynamics simulation analyses. The putative laccase CB10_180.4889 (Lac_CB10), composed of 728 amino acids, with theoretical molecular mass values of approximately 84 kDa and a pI of 6.51, was predicted to be a new CopA with three cupredoxin domains and four conserved motifs linking MCOs to copper sites that assist in catalytic reactions. Molecular docking studies revealed that Lac_CB10 had a high affinity for the molecules evaluated, and the affinity profiles with multiple catalytic pockets predicted the following order of decreasing thermodynamically favorable values: tetracycline (−8 kcal/mol) > ABTS (−6.9 kcal/mol) > sulfisoxazole (−6.7 kcal/mol) > benzidine (−6.4 kcal/mol) > trimethoprim (−6.1 kcal/mol) > 2,4-dichlorophenol (−5.9 kcal/mol) mol. Finally, the molecular dynamics analysis suggests that Lac_CB10 is more likely to be effective against sulfisoxazole-like compounds, as the sulfisoxazole-Lac_CB10 complex exhibited RMSD values lower than 0.2 nm, and sulfisoxazole remained bound to the binding site for the entire 100 ns evaluation period. These findings corroborate that LacCB10 has a high potential for the bioremediation of this molecule.

## 1. Introduction

Laccases ((benzenediol: oxygen reductases (EC 1.10.3.2)) belong to the group of multicopper oxidases (MCOs), together with other enzymes, such as ascorbate oxidase, ferroxidase and bilirubin oxidase [1]. MCOs are enzymes that have four copper atoms per monomer, which are essential for the correct catalytic performance of the enzyme. These atoms are divided into centers that differ by paramagnetic resonance signals, called type 1 copper (T1, blue copper) centers, type 2 copper (T2, normal copper) centers and type 3 copper (T3, coupled binuclear) centers [2].

These enzymes can be found in plants, fungi, bacteria [3] and insects [4], and their function is related to their organism of origin and environmental conditions. Fungal laccases are known for their participation in pigment production, morphogenesis, sporulation and degradation of lignocellulosic material [5]. Due to their broad spectrum of action on various substrates, bacterial laccases have a wide range of biotechnological applications, including the bleaching of paper and cellulose, the discoloration and degradation of dyes in textile effluents [6], and the degradation of antibiotics, such as flavonoids and phytoalexins [7].

The use of bacterial laccases in biotechnological applications has been growing compared to that of fungal laccases due to several advantages, such as their ability to function in a wide range of temperatures and pH values [8]. This characteristic is very important for industrial processes, since most such processes occur under extreme conditions. Another factor of interest is the relatively low production cost of bacterial laccases, considering the rapid growth and enzyme production rates of bacteria, as well as the relative ease of cloning and expression in heterologous hosts when compared to cloning in yeast or mammalian cells [9,10].

One current concern is the persistent accumulation of residues generated through the recurrent use of antimicrobial compounds for human and veterinary use, as well as synthetic compounds used as pesticides in agriculture and as inputs in other industrial sectors. Antibiotics such as tetracyclines, sulfonamides and trimethoprines are often detected in effluent from water treatment plants [11], indicating that traditional treatment methods are not effective for the removal of various chemical components and their potentially more toxic metabolites present in effluents, significantly increasing the selection of resistant bacteria. Another concern is the bioaccumulation of these components along the food chain, including in humans, due to the consumption of contaminated fish, crustaceans and vegetables [12].

In the agricultural sector, the synthetic pesticide 2,4-dichlorophenol (2,4-DCP) has been consistently found in effluents due to the improper disposal of this phenolic compound, which can present serious risks to human and environmental health [13].

Benzidine is a synthetic compound widely used in the production of dyes applied in the textile, paper and leather sectors. The improper disposal of this substance in industrial effluents can present serious risks to human and environmental health, considering that this substance is reported as a carcinogen by the United States Department of Health and Human Services and the International Agency for Research on Cancer [14].

Bacterial laccases have great biotechnological potential for the degradation of the above compounds because they can degrade several recalcitrant chemical structures, including complex phenolic radicals. No reported study has used molecular docking simulation to investigate the interaction of a laccase of *Chitinophaga* sp. with these substances associated with great environmental and public health problems. Therefore, this study aimed to evaluate the potential of a laccase recovered using a sequence-driven prospecting strategy from the Laboratory of Biochemistry of Microorganisms and Plants (LBMP) genomic database. Our laccase was isolated from a biomass-degrading bacterial consortium of *Chitinophaga* sp. CB10, a Gram-negative, non-pathogenic, filamentous, mesophilic, immobile, non-spore-forming bacterium [15]. This microorganism has been shown to be extremely promising source of enzymes of industrial and biotechnological interest [15,16,17,18,19,20].

In the present study, in silico approaches were used to understand the general parameters of the enzyme, analyze its conserved regions and tertiary structure, and assess its interaction with compounds of environmental importance by the molecular docking technique. Furthermore, molecular dynamics simulations were performed to observe important conformational changes of Lac_CB10 in the protein–ligand complex.

## 2. Results and Discussion

### 2.1. Screening and Functional Prediction

The search for laccases based on the conserved L3 domain (H-P-M-H-L-H-G-H) in the internal database of the LBMP returned 43 potential sequences for laccase, which were subjected to a new screening by Pfam, resulting in only one open reading frame (ORF) containing the three complete cupredoxin domains: Lac_CB10.

To draw a homology profile for Lac_CB10, the similarity between the ORF and other sequences deposited in the GenBank public database of NCBI was analyzed using the BLASTp tool. The collection of nonredundant sequences (nr) was selected: nr is a comprehensive database that contains all sequences deposited in GenBank, although it is not manually curated. Lac_CB10 shared 96% identity with a hypothetical protein of *Chitinophaga jiangningensis* (accession number WP_073081050.1), in addition to showing similarity (96.16% to 63.39%) to other sequences of *Chitinophaga* sp. This finding indicated that it is an ORF found in the genomes of this genus, without evidence of assembly and annotation artifacts.

To verify the percent similarity to other sequences in the functional prediction, Lac_CB10 was compared using the BLASTp tool against the UniProtKB/SwissProt collection of sequences, a manually curated database. The comparison predicted 33% identity with a CopA, with high tolerance to copper (Q.47452.1) associated with *Pseudomonas syringae*, corroborating the prediction that Lac_CB10 is a laccase. Furthermore, the results indicated that there is no previously experimentally characterized sequence with a high identity with the target sequence. Similar sequence novelty was obtained using the Protein Data Base (PDB) collection, which contains sequences that have been structurally studied using experimental crystallographic techniques; Lac_CB10 shared only 33% (46% query coverage) identity with a prospected laccase from a fungus, *Botrytis aclada* (PDB ID 3V9E).

Interestingly, the highest identity result from the PDB collection was a fungal laccase, demonstrating a lack of structural and functional studies on bacterial laccases, as well as a historical preference for the use of enzymes of fungal origin in industrial processes; this bias is especially evident in the designation of certain enzymes, generally recognized as safe after decades of use. However, according to Christopher et al. [21], bacterial laccases have advantages for industry because they have broad substrate specificity and allow for rapid enzyme production.

Considering that the low-identity results obtained against the SwissProt (33%) and PDB (33%) data collections may indicate either an unprecedented sequence or a sequence protected by a patent, similarity analysis was also performed against the patent collection of GenBank (pataa). The results indicated that the study sequence is not the subject of any patent claim, considering that the maximum result was 50% identity and 64% coverage, shared with patent US9290773 (accession number APN18218.1), which refers to transgenic plants with recombinant DNA for the expression of proteins that confer greater water use efficiency, greater tolerance to cold, increased production yield and increased use of nitrogen. Appendix A presents the main results for the identity analysis of the chosen laccase in relation to sequences deposited in the NCBI GenBank public sequence database. According to the data obtained from the more reliable curated database (SwissProt), the analyzed sequence has the greatest similarity with laccase sequences of the CopA superfamily.

Predicting the physicochemical parameters of a protein, such as the isoelectric point, molecular weight, and instability index, is the first step to analyze the unique properties of the molecule [22]. The in silico study of these parameters is of great importance to select target molecules for in vitro experimental studies and assists in the steps that would be extremely time-consuming if they were optimized only by empirical approaches, especially extraction and purification steps. The results are shown in Appendix A. Lac_CB10 is predicted to be stable, with a half-life of 10 h in *Escherichia coli* as a heterologous expression model, in addition to having good thermostability.

Regarding the chemical composition of proteins, the five most common amino acids in Lac_CB10 were leucine (Leu, 8.1%), glycine (Gly, 8%), threonine (Thr, 6.9%), valine (Val, 6.7%) and asparagine (Asp, 6.5%), as shown in Figure 1A. Leucine (Leu) and valine (Val) are both amino acids with aliphatic side chains, which have hydrophobic and nonpolar groups formed by a carbon chain with only single bonds. The aliphatic nature of these side chains contributes to the stability of Leu and Val molecules in a number of ways, such as allowing them to interact favorably with other hydrophobic groups within a protein, and can lead to the formation of hydrophobic clusters that help stabilize the tertiary structure of the protein [23]. Furthermore, the absence of reactive functional groups on the aliphatic side chains of Leu and Val can help prevent unwanted chemical reactions from occurring within a protein. This may further contribute to the stability of the protein structure [24]. In general, the aliphatic side chains of Leu and Val are important for the stability and function of many proteins. They play a critical role in protein folding and stability, and are involved in a variety of protein–protein interactions [25].

MCOs are characterized by the presence of two or three cupredoxin domains. These domains are important for the catalytic performance of the enzyme since they contain the highly conserved histidine and cysteine residues that coordinate copper atoms [25], as well as the substrate/enzyme binding site [26].

According to the Laccase and Multicopper Oxidase Engineering Database (LccED) [27,28], MCOs are grouped into 105 homologous families and 16 superfamilies, of which 11 contain enzymes from three cupredoxin domains, and only 1 superfamily is associated with six domains. Of the 16 superfamilies, 8 are mainly eukaryotic organisms, including fungi, plants, and insects; 7 are bacteria; and only 1 superfamily, comprising two domains, is archaea.

To confirm the homology of Lac_CB10 with the group of MCOs, the protein sequence predicted for this ORF was subjected to analysis of conserved domains to identify cupredoxin domains using the Pfam platform. The results indicated the presence of three cupredoxin domains, namely, “Cu-oxidase 3”, “Cu-oxidase” and “Cu-oxidase 2” (Figure 1B), composed of 107, 81 and 112 amino acids, respectively. Within the MCOs, the superfamily of copper resistance protein A (CopA) is monomeric and contains three cupredoxin domains in its structure. Although having three domains is not an exclusive characteristic of this superfamily, the prediction that Lac_CB10 is a CopA is supported by the Pfam analysis of conserved domains, to the extent that the analysis excludes Lac_CB10 from the groups characterized by two domains, such as small laccases (SLACs) [29].

To investigate which superfamily Lac_CB10 should be grouped into based on a distance matrix, a phylogenetic tree was constructed by neighbor-joining with the seven bacterial MCO superfamilies obtained from the following laccase and MCO databases: Biocatnet LccED v6.4 [30,31]: H—Bacterial CopA; I—Bacterial Bilirubin Oxidase; J—Bacterial CueO; L—Bacterial MCO; K—SLAC-like (type B 2dMCO); N—Bacterial type B 2dMCO and O—Archaeal and Bacterial type C 2dMCO. Figure 2 shows the phylogenetic tree with the clades colored according to their expected classification among the superfamilies of the LccED database, which validates the chosen clustering method (phylogenetic cladogram obtained for the distance matrix optimized by the neighbor-joining method). Appendix A shows the sequence ID information of the analyzed organisms.

The prediction of homology between Lac_CB10 and laccases of the CopA group (Appendix A) corroborated with the phylogenetic data, so Lac_CB10 was grouped as a sister group of the superfamily clade H—CopA with 91% support (Figure 2). These findings are extremely relevant because the CopA enzyme superfamily is potentially involved in increased resistance to copper, which makes these enzymes important for the treatment of effluents and contaminated soils, in addition to all other biotechnological applications of laccases. The laccases belonging to this superfamily also exhibit mechanisms for dye bioremediation, such as the laccase of *Stenotrophomonas maltophilia* AAP56, which is involved in dye-bleaching in vitro [32].

The conserved patterns L1 (H-W-H-G) and L3 (H-P-x-H-L-H-G-H) are specific for laccases and contain ligands for the copper centers. The L1 pattern has a histidine that binds to the T2 copper site and a histidine that binds to the T3 site, while the L3 sequence has ligands at all three copper sites (T1, T2 and T3). In the H family (Bacterial CopA), which presented the best homology results for Lac_CB10, 98% of the sequences are of bacterial origin and 83% are recorded as laccases in GenBank; among these laccases, 50% have the L1 signature and 42% have the L3 signature [28].

In addition to the conserved motifs L1 and L3, laccases have two other conserved motifs that are directly involved in binding at copper sites, namely L2 (H-X-H) and L4 (H-C-H-X-X-X-H-X-X-X-X-M/L/F) [33]. The knowledge of the motifs conserved in known enzymes assists in the search and prospection of targets in sequenced genomes, and metagenome motifs are used as an initial screening criterion to search for enzymes of interest since enzymes from other classes are already discarded in the first stage of screening.

Considering the high similarity with CopA, multiple alignment was performed with five sequences of class H laccase (CopA laccase) (Figure 3) to identify the four conserved motifs, L1 (H-W-H-G), L2 (H-X-H), L3 (H-P-x-H-L-H-G-H) and L4 (H-C-H-X-X-X-H-X-X-X-X-M/L/F), that are responsible for the coordination of the three copper centers. In addition, the alignment showed a high conservation index among several blocks in the sequences, once again reinforcing that the enzyme is in this group.

The secondary structure analyses indicate that Lac_CB10 consists mainly of coils, followed by beta strands, alpha helices and beta turns, as shown in Figure 4. Studies with fungal and bacterial laccases have shown that in all the enzymes studied, the most predominant structure was also coils [34]. Coils play an important role in protein flexibility, conformational changes, and enzymatic turnover [35]. These characteristics confer broad structural plasticity and may confer a greater capacity for the adaptation and expansion of enzyme affinity to various types of substrates. This broad action on different substrates with different degrees of affinity and specificity is common to laccases reported in the literature [6,7]. Such properties have promoted high interest in the application of the enzyme in several biotechnological research fields, such as biosensors for the detection of phenols in wastewater, delignification to produce second-generation ethanol, bleaching of cellulose for paper production and the bioremediation of recalcitrant compounds, such as antibiotics and dyes released in effluents [36].

Thus, based on the similarity results with public database sequences, localization of conserved domains for cupredoxin, the identification of specific signatures and secondary structure composition, we can infer that Lac_CB10 has relevant homology results that indicate a strong probability of classification of this enzyme as a laccase, with relevant evidence of it belonging to the CopA superfamily.

### 2.2. Modeling and Tertiary Structure

To obtain information on the structural conformation of Lac_CB10, a three-dimensional model was inferred by homology (Figure 5). The generated model (template PDB ID 3pps with 30.84% identity) was modeled based on the most similar template deposited in the PDB for the fungal laccase of *Canariomyces arenarius*. Among the templates of bacterial laccase structures, the similarity was very low, demonstrating that there is a shortage of bacterial laccase structures deposited in the PDB; this shows the importance of further studies on this class, since the minimum identity considered necessary for the construction of a reliable protein structure prediction model is 30% [37].

The obtained model predicted a monomeric three-dimensional protein structure with three cupredoxin domains, as expected for most MCOs, including CopA [38]. Since a fungal structure model was selected, 30% similarity fell within the conserved cupredoxin domains, where the highest rates of local similarity between the template and model were obtained for Cu-oxidase 3 and Cu-oxidase, followed by Cu-oxidase 2 (Appendix A).

The quality of the model was evaluated by the Ramachandran plot server [39]. The Ramachandran plot analysis (Figure 5B) of the model showed that 96.14% of the residuals were in favorable regions, indicating that the 30.84% similarity of the Lac_CB10 structure model with the template was sufficient for further in silico molecular docking analyses.

As previously highlighted, laccases have three copper-binding centers, T1, T2 and T3; T3 centers are also called binuclear centers because they have two copper molecules, and T2 and T3 centers positioned in the shape of a triangle are called T2/T3 copper trinuclear centers. These copper atoms are coordinated by a methionine residue, a cysteine residue and 10 highly conserved histidine residues that are considered essential for the correct catalytic performance of these macromolecules. These histidines are located within the L1 (H-W-H-G), L2 (H-X-H), L3 (H-P-x-H-L-H-G-H) and L4 (H-C-H-X-X-X-H-X-X-X-X-M/L/F) signatures [38].

The generated model has four conserved signatures that are related to the coordination of the copper centers, but the automatic modeling method using SWISS-MODEL recognized only the T2/T3 trinuclear center, and only three copper atoms were included as ligands in the obtained model. However, given the known high conservation of the four copper atoms in laccase structures previously described in the literature and knowing that the Lac_CB10 model contained all coordination sites for these ligands, a copper atom corresponding to the T1 center was added manually. For this purpose, an alignment was performed with the Blast tool using the amino acid sequence of Lac_CB10 against the PDB database of crystallographic structures, where structures with a high percentage of coverage (70%) were chosen. Next, these structures were aligned with Lac_CB10 by the “Alignment” plugin in the PyMOL tool to determine whether the copper atoms and conserved motifs overlapped correctly. After alignment, the structure (PDB ID 1V10, *Rigidoporus lignosus* laccase) that showed the best overlap was used for the addition of the copper atom T1, with 70% coverage and an RMSD value of 2.175 Å.

Figure 6 shows the four copper atoms, represented by orange spheres, and the four conserved motifs that coordinate these atoms and their respective residues. The L1 motif (H-W-H-G) is colored yellow, the L2 motif (H-S-H) is colored purple, the L3 motif (H-P-M-H-L-H-G-H) is colored blue and the L4 motif (H-C-H-I-L-Y-H-M-M-S-G-M) is colored pink. Studies have suggested that the PHE460 and ILE452 residues near the T1 copper center favor a high redox potential between the ligand and enzyme [4], and that laccases with a PHE residue as the axial ligand of the T1 copper have a high redox potential, while laccases with a MET residue have a low redox potential [40,41]. Lac_CB10 showed an ILE516 hydrophobic residue near the T1 center, and the copper site was coordinated by a cysteine and two histidines (Appendix A), demonstrating that the interaction of PHE or MET residues as axial ligands of the T1 site does not universally influence the redox potential in all laccases. A similar observation was reported in another recent study that analyzed 14 gene sequences of laccases identified in the genome of the fungus *Amylostereum areolatum* [35].

From the homology modeling results, the fact that our three-dimensional model of Lac_CB10 was inferred from a fungal laccase template reinforces the scarcity of bacterial laccase structures that have been experimentally characterized by crystallographic techniques, which reinforces the need for further studies to elucidate structures, in silico or experimental. These same results also demonstrate the conservation of copper-binding domains among bacterial and fungal laccases. All four conserved motifs (L1–L4) responsible for the binding metal cofactors, specifically the copper ions involved in the enzyme’s catalytic mechanism, were found in the sequence.

The best similarity results were observed within the three conserved cupredoxin domains (Appendix A). This allowed for the predicted structure model obtained in this study to be used for molecular docking analysis.

Despite the inherent limitations of in silico predictive studies and considering that the similarity between the template and model reached the recommended minimum for in silico studies of homology modeling (~31%), this study is relevant. It contributes to improving our understanding of the action mechanisms of bacterial laccases and provides evidence for future crystallographic studies that will expand our knowledge of the catalytic diversity of enzymes such as laccases.

### 2.3. Molecular Docking and Ligand Preparation

Human knowledge of antimicrobials has evolved significantly and has generated benefits for modern society; however, these benefits come with a new problem: bacteria resistant to these drugs. One of the main routes for the emergence of these pathogens is selective pressures exerted by antibiotic residues [42]. These residues have been reported to affect the microbial populations involved in anaerobic digestion in the soil, causing low digestion efficiency and consequently a low nutrient recycling rate [43]. Considering the impact of these residues on the environment, as well as their effect on the emergence of multidrug-resistant pathogens, studies focused on the search for enzymes capable of the bioremediation of environments contaminated with these persistent molecules are extremely important. In this context, laccases exhibit a high potential for biotechnological application in the degradation of these compounds and many others, since they can degrade several recalcitrant chemical structures, including complex phenolic radicals.

Therefore, we performed molecular docking assays to study how molecules of environmental interest would be docked in the Lac_CB10 laccase cavity to elucidate the ligand–protein interaction patterns and the potential of Lac_CB10 enzyme for the degradation of 2,4-dichlorophenol, sulfisoxazole, tetracycline, benzidine and trimethoprim (Appendix A). Assays were performed to predict which ligands could be the potential substrates of this new laccase based on the best positioning of the enzyme–ligand interaction, which was determined through the simulation of the Gibbs free energy (kcal/mol) associated with contact between the enzyme and ligand upon docking of the ligand in the catalytic pocket. Laccases in general have high specificity for ABTS, which was used as a control in this study. ABTS is used as a redox mediator to increase the efficiency of the reaction between the enzyme and ligands, since this synthetic compound can produce electroactive species that facilitate substrate oxidation and mono- and dicationic species that help continue the reaction in the absence of the enzyme [44]. Gibbs free energy analysis can predict enzyme–ligand catalytic interactions because when the enzyme–ligand complex has a free energy greater than 0, the affinity of the complex is low, but the more negative this energy is, the greater the affinity of the bond.

For these analyses, the search grid was defined to cover the entire Lac_CB10 molecule. This approach allowed for the ligands to be docked to the catalytic cavity with a high affinity and did not force them to bind only to a specific region that had been previously selected. All ligands in this study showed an interaction with catalytic cavities of the enzyme, and based on the Gibbs free energy values, tetracycline had the highest binding affinity, followed by ABTS, sulfisoxazole, benzidine, trimethoprim and 2,4-dichlorophenol. The affinity (kcal/mol) and RMSD (Å) results are shown in Table 1.

The obtained enzyme–ligand representations are shown in Figure 7A–F, and the schemes produced using the LigPlot program are shown in Appendix A.

As expected, the laccase–ABTS complex showed a high affinity, with a binding energy of −6.9 kcal/mol and an RMSD of 0 Å. ABTS interacted hydrophobically with residues Gln115, Gly267, Met371, Ala268, Asp269, Pro148 and Ile147, and formed hydrogen bonds with residues Val114, Lys150, Glu370 and Val151 (Figure 7A).

Tetracycline, a broad-spectrum antimicrobial of Gram-positive and Gram-negative bacteria, did not interact with any residues belonging to the copper-binding sites. Nevertheless, it showed the strongest interaction (−8 kcal/mol) among not only the class of antimicrobials, but also all the compounds tested, including ABTS. The efficiency of laccases in tetracycline degradation is very high in in vivo experiments, as demonstrated in the study of Yang, J et al. [45], where 40 U/mL immobilized *Cerrena unicolor* laccase was able to remove 100 mg/mL tetracycline at pH 6.0 and room temperature without the aid of a redox mediator. Tetracycline interacted hydrophobically with residues Leu477, Val476, Thr372, Tyr368 and Pro369, and formed hydrogen bonds with residues Arg475, Thr365, Thr291 and Met367 (Figure 7B).

Within the category of antibiotics, sulfisoxazole, a broad-spectrum sulfonamide antibiotic, exhibited the second highest affinity (−6.7 kcal/mol) and interacted hydrophobically with the residues belonging to two conserved copper-binding motifs, Cu-II (His 124) and Cu-IV (His 513 and Met 521), as well as with Gly 525, Phe 511, Leu 410, His 513, His 124, Met 250, Met 126, Met 521, Tyr 95, Pro 411 and Lys 412 (Figure 7C).

The antimicrobial trimethoprim showed an affinity of −6.1 kcal/mol and interacted hydrophobically with His124, which belongs to the conserved motif bound to CuII, Lys412, Met521 (CuIV), Leu410, Pro411 and Tyr95, and established hydrogen bonds with Met520 (CuIV) and Met126 (Figure 7D). The interaction of trimethoprim with CuIV (His479) was also described by Cárdenas-Moreno et al. [44], who reported a binding energy of −6.5 kcal/mol, very close to our result.

Phenolic compounds have been widely used to produce dyes, drugs and pesticides, and are coproduced as waste in many processes, such as paper production. When inadvertently discarded in effluents, these substances cause severe environmental harm, including detrimental effects on human and animal health [46]. Among these compounds, benzidine, a carcinogenic synthetic compound, is widely used in the manufacture of dyes, and treating residues of this compound is of extreme environmental importance. Similar to benzidine, 2,4-dichlorophenol (2,4-DCP) is a synthetic phenolic compound extensively utilized in pesticides. It is also found in water bodies due to unintentional disposal and leaching, which is linked to the compound’s persistence. Both benzidine and 2,4-dichlorophenol are chemical compounds employed in diverse industrial applications. They are recognized as toxic and can potentially endanger human health and the environment if not handled or disposed of correctly [13].

Similar to the ligands mentioned above, benzidine (−6.4 kcal/mol) also forms a hydrophobic interaction with the residues present in the CuII (His124) and CuIV (His513, Met520, Met521) motifs, and with Phe511, Leu410, Tyr95, Lys412 and Met126 (Figure 7E). The laccase-2,4-DCP complex showed the weakest interaction (−5.9 kcal/mol). Unlike the ligands mentioned above, except tetracycline and ABTS, 2,4-DCP binds to only one motif of the copper sites, CuII (His 124), a residue that bonds hydrogen and interacts hydrophobically with Phe 511, Tyr 95, Leu 410 and Lys 412 (Figure 7F).

ABTS was used as a control because it is used as a redox mediator and represents the substrate with the greatest affinity for laccase. Spore coat protein A (CotA), a laccase isolated from *Bacillus subtilis*, is used as a model to study both structure and function, and it has been reported that the ABTS binding site is close to the T1 center [47]. Interestingly, the binding site of our Lac_CB10 to ABTS was located far from the T1 center, approximately 27 Å (Appendix A), indicating a possible new catalytic pocket in our bacterial laccase. Liu et al. [48] also identified new catalytic sites for laccases and reported a distant ABTS-binding site located 26 Å away from the T1 center.

All ligands, except ABTS and tetracycline, interacted with copper-binding sites Cu-II and Cu-IV; among these interactions, all compounds interacted with His 124 (Cu-II): sulfisoxazole interacted with His 513 and Met 521 (Cu-IV); trimethoprim interacted with Met 520 and Met 521 (Cu-IV); benzidine interacted with His 513, Met 520 and Met 521 (Cu-IV); and 2,4-DCP interacted only with His 124 (Cu-II). Tetracycline did not interact with any of the conserved motifs. This result was consistent with the findings of a study by Cárdenas-Moreno et al. [44] that also analyzed the laccase–tetracycline interaction, reporting that there was no interaction with the conserved residues of the copper sites, but also showed a high affinity (−5.2 kcal/mol). This lack of interaction between the residues may have occurred because tetracycline has many polar groups that are unable to interact with the amino acids present in the copper sites [44].

This flexibility, where each ligand has unique relationships to the amino acid residues and catalytic pockets with higher affinity, is supported in the literature; it was previously believed that the catalytic efficiency of laccases was correlated with the difference in redox potential between the substrate and the T1 copper center [49]. However, this statement has since been strongly contested by studies reporting that the catalytic efficiency of laccases depends not only on the difference in redox potential between the substrate and the copper center, but also on other physicochemical characteristics of the substrate itself, such as the steric characteristics and the conformation of the substrate and connection pockets [50,51]. In our Lac_CB10, three binding pockets were found: ABTS and tetracycline were in different pockets, and sulfisoxazole, trimethoprim, benzidine and 2,4-DCP were docked in the same pocket, even sharing some common residues. The four ligands that are docked in the same pocket are structurally similar, with two aromatic rings each. Moreover, 2-4-DCP is the smallest ligand studied, with only one aromatic ring, and forms hydrogen bonds with only one residue, which also interacts with the other ligands, except for ABTS and tetracycline.

This flexibility of the enzyme in terms of efficiency, which is dependent not only on the catalytic pockets of the enzyme itself, but also on substrate factors, makes laccase enzymes appropriate for the bioremediation of different compounds. Table 2 shows the ligands and their respective interaction residues

We conducted dynamic simulations to study the variation in the molecular behavior of the complexes. The corresponding trajectories were used to conduct various evaluative procedures to analyze structural insights and stability. These included the root mean square deviation (RMSD), root mean square fluctuation (RMSF), radius of gyration (Rg) and solvent-accessible surface area (SASA).

Upon analyzing the RMSD of the ligands, we observed that only the sulfisoxazole (named sulfiso) complex ligand displayed RMSD values smaller than 0.2 nm among the enzymes analyzed (Figure 8). Trajectory frames demonstrated that the ligand remained bound to the binding site in the sulfiso–Lac-CB10 complex.

To evaluate the stability of each system, we analyzed the RMSD to measure the average distance between the atoms of each protein concerning the frames obtained during the simulation. Figure 9 shows the RMSD of Lac-CB10 complexed with sulfiso, revealing that the trajectory of the enzyme in the interaction with the ligand tends to stabilize during the simulation.

To assess which amino acid residues contribute to trajectory fluctuations, we analyzed the RMSF of the sulfiso–Lac-CB10 complex (Figure 10). We noticed that all the residues present in the fluctuations are part of the cupredoxin Cu-oxidase 3 domain at the N-terminus of the enzyme and are involved in the formation of secondary structures such as beta strands and loops. The beta sheet structures are maintained by hydrogen bonds, providing rigidity and stability, and the loops are more flexible regions, without a defined structure, but with an important role in protein stabilization, as they allow the protein to adapt to different environments [8]. However, the third fluctuation peak draws the most attention, as it contains amino acid residues that make up the conserved LI motif (H-W-H-G), which interacts directly with the copper atoms of the trinuclear cluster T2/T3, showing that it is related to the binding of the cofactor and substrate [52], in addition to direct participating in the substrate redox reaction. These results indicate that when the flexibility of an enzyme decreases, conformational changes may occur in relation to substrate binding so that the catalytic pocket adapts to achieve a perfect fit to the ligand [53].

To determine the overall compaction of the protein that remained bound to its ligand, we calculated the radius of gyration (Rg) (Figure 11). The interaction with sulfiso displays slight compression and decompression movements; however, the trajectory tends to reach equilibrium at approximately 2.6 nm.

The analysis of the solvent-accessible surface area (SASA) describes the portion of a protein’s molecular surface that is accessible to the solvent. The trajectories of the sulfiso–Lac-CB10 complex stabilized at approximately 320 nm^2^ (Figure 12).

The low fluctuation and the small RMSD shift demonstrate that the enzyme–linker complex is stable for the bioremediation of the compound sulfisoxazole. In addition, the results show that the MDam simulations remain stable throughout the 100 ns reaction.

## 3. Materials and Methods

### 3.1. Screening and Functional Prediction

The initial screening of genomes and metagenomes in the internal collection of LBMP was performed by searching for one of the known L3 laccase signatures (H-P-M-H-L-H-G-H), and differentiating it from others as described in the literature [27,28]. The sequences that had the L3 pattern were analyzed using the Pfam tool [54] to verify the presence of motifs related to the family of conserved domains, and the sequences with complete domains belonging to the group of MCOs were selected.

The degree of similarity shared between the selected sequences and enzymes characterized in the American database of the National Center for Biotechnology Information (NCBI) (collection “Non-redundant UniProtKB/SwissProt sequences”) was also investigated using the Basic Local Alignment Search Tool (BLAST) [55], as was the similarity with patented sequences from this same database (patented protein sequence collection (pataa)).

The best hit corresponded to the amino acid sequence of Lac_CB10 in the genome of *Chitinophaga* sp. CB10, which was isolated from a consortium (SisGen—National System for the Management of Genetic Heritage and Associated Traditional Knowledge—number A390589) of a sugarcane bagasse pile, with a high rate of biomass degradation that was cultivated with carboxymethylcellulose as the only carbon source [18]. The genome of this microorganism was partially sequenced [15], has been deposited under accession number MLAV00000000 in NCBI GenBank and is part of the internal collection of genomes and metagenomes of LBMP.

For the in silico characterization, analyses of the physicochemical parameters of the predicted bacterial laccase were performed using the ExPASy-ProtParam tool [22], which allows for the visualization of the composition and abundance of amino acid residues, molecular mass, isoelectric point, molar extinction coefficient, instability and aliphatic indices, and grand average of hydropathicity (GRAVY). The analysis of the secondary structure was performed using the SOPMA tool [56] and PREDiction (PSIPRED) [40] to predict the percentage of secondary structures in alpha helices, beta sheets and random coils.

The phylogenetic dendrogram was generated by the program MEGA7 (Molecular Evolutionary Genetics Analysis version 7.0) for larger datasets [41], based on the distance matrix generated from an alignment obtained by the BioEdit Sequence Alignment Editor [57]. The neighbor-joining clustering optimization method was used, and bootstrapping sampling of 5000 replicates and the Poisson substitution model were employed. The tree was edited using the program FigTree [30].

Five sequences of each MCO superfamily found in bacteria were used, namely, Bacterial CopA—H; Bacterial Bilirubin Oxidase—I; Bacterial Cueo—J; Bacterial MCO—L; SLAC-like (type B)—K; Bacterial type B 2dMCO—N; and Archaeal and Bacterial type C—O. The sequences were obtained from the database of laccases and MCOs in Biocatnet LccED v6.4 [27,28] (Appendix A). The alignment to demonstrate the conserved patterns was performed using the online software Praline [31], with five sequences belonging to the CopA family obtained from the database of laccases and MCOs in biocatnet LccED v6.4.

### 3.2. Modeling by Homology and Tertiary Structure

Structural modeling by homology was performed using the Swiss-Model online platform [58]; eight homologous models were generated, from which the model with the highest percentage of similarity and prediction quality (best Qmean result) was chosen. The selected model was visualized and edited using the program PyMOL (version 4.3) [59].

### 3.3. Molecular Docking and Ligand Preparation

The molecular docking simulation was performed to study the interaction of Lac_CB10 with various types of ligands. For this purpose, the program AutoDock Tools 1.5.7 [60] was used to prepare the receptor molecule; the water molecules were removed, polar hydrogen atoms were added, and the coordinates of the search grid were established. The search grid size parameters were optimized, and the best molecular interaction results according to the binding energy were obtained using the total structure of the enzyme, not only the catalytic site residues.

The coordinates of the search grid were x = 66, y = 62 and z = 82 with 1 Å spacing; the center of the grid was located at x = 18.833, y = 0.028, z = 9.833; and the exhaustiveness parameter was 10. Molecular docking analysis was performed with AutoDock Vina 1.1.2 [61]. The best binding mode for each ligand was the mode with the lowest free binding energy, which was aligned with the receptor structure and chosen for subsequent analysis. The visualization of the protein–ligand structure and the preparation of the figures were performed with the PyMOL software (version 4.3) [59].

The structures of the following ligands were obtained from the PubChem Chemical Structure Search database [62]: 2,4-dichlorophenol, 2,2′-azino-bis (3-ethylbenzthiazoline-6-sulfonic acid) (ABTS), benzidine, sulfisoxazole, tetracycline and trimethoprim. The ligand molecules were modeled for flexibility in the program AutoDock Tools 1.5.7 [60], with the number of twists adjusted to the maximum value for each ligand.

### 3.4. Molecular Dynamics Simulations

GROMACS package version 2019.3 [63] was used for the molecular dynamics simulations of the complexes. The force field used was CHARMM36 [64]. The addition of hydrogens in each ligand, considering the protonation state of the atoms at pH 7.4, was performed using the Avogadro software version 1.2.0 [65]. The ligand parameters were obtained from the CGennFF server [66]. The complexes were centered in the box and placed 10 Å from the box edge, and the molecules were solvated in a cubic box with TIP3P water molecules and neutralized by adding the appropriate number of Na^+^ Cl^−^ ions, considering an ionic concentration of 0.15 M. Energy minimization was carried out using the steepest descent method with a maximum force of 1000 Kj/mol.nm for 50,000 steps. After minimization, the system was equilibrated in two steps: a canonical NVT (number of particles, volume, and temperature) ensemble followed by an isothermal–isobaric NPT (number of particles, pressure and temperature) ensemble. NVT equilibration was carried out at a constant temperature of 300 K for 500 ps, while NPT equilibration was performed at a constant pressure of one bar and constant temperature of 300 K for 500 ps. The production phase was performed at 300 K for 100 ns. The integration time step for all the simulations was set to 2 fs, and trajectories were saved every 0.50 ns.

## 4. Conclusions

The putative laccase CB10_180.4889 (Lac_CB10), composed of 728 amino acids, with a theoretical molecular mass of approximately 84 kD and a pI of 6.51, was predicted as a new CopA, with three cupredoxin domains and four motifs conserved for specific MCOs for laccase (L1-L4) bound to copper sites that aid in catalytic reactions. As revealed by molecular docking studies, Lac_CB10 showed a high affinity for the evaluated molecules, presenting predicted affinity profiles with multiple catalytic pockets in the following order of decreasingly thermodynamically favorable values: tetracycline (−8 kcal/mol) > ABTS (−6.9 kcal/mol) > sulfisoxazole (−6.7 kcal/mol) > benzidine (−6.4 kcal/mol) > trimethoprim (−6.1 kcal/mol) > 2,4-DCP (−5.9 kcal/mol). This multiplicity of catalytic pockets in the same enzyme demonstrates the importance of not limiting the selection grid to only one catalytic pocket, but rather extending the selection box to the entire enzyme so that the ligands fit in the pockets with the greatest affinity. The results obtained from the MD analyses demonstrate that the Lac_CB10–sulfisoxasole complex was the only one that maintained its stable trajectory throughout the entire time of the analysis, indicating that this enzyme–ligand complex is a possible candidate for the bioremediation of this compound. The predicted model of the tertiary structure of Lac_CB10 based on homology shows that there are few crystallographically resolved structures of bacterial laccases deposited in the relevant databases, which highlights the importance of this work. The present results represent a relevant contribution to bioinformatic research on bacterial laccases, since most laccases deposited in databases are of fungal origin. These findings will assist in basic research to elucidate the physicochemical and catalytic properties of prokaryotic laccases. Furthermore, we strongly advocate for conducting additional biochemical characterization studies to validate these findings through in vitro assays, thereby generating promising research avenues concerning the involvement of bacterial laccases in environmental bioremediation processes.

## Figures and Tables

**Figure 1 ijms-24-09785-f001:**
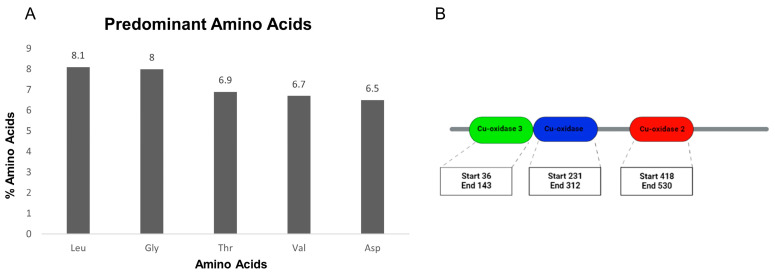
Amino acid abundance analysis and conservation of typical MCO/laccase domains. (**A**) Bar graph representing the five most predominant amino acids in Lac_CB10; (**B**) scheme of the Cu-oxidase 3, Cu-oxidase and Cu-oxidase 2 domains belonging to Lac_CB10, showing the amino acid numbers at which the domain starts (start) and ends (end).

**Figure 2 ijms-24-09785-f002:**
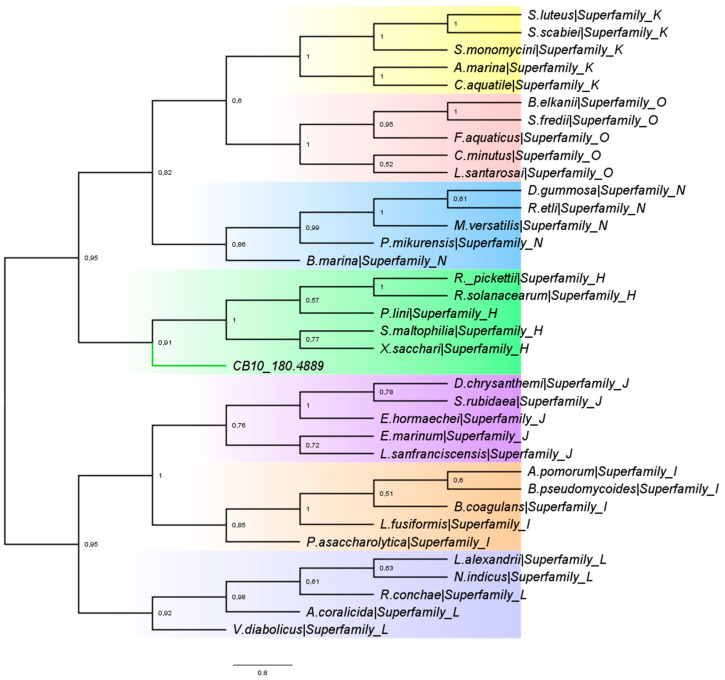
Phylogenetic dendrogram of Lac_CB10 against the laccase classes deposited in the laccase and MCO database (“LccED”) based on the distance matrix generated from an alignment obtained using the Mega 7 program by neighboring-clustering optimization. The yellow arm indicates the K superfamily, the pink arm indicates the O superfamily, the blue arm the N superfamily, the purple arm the J superfamily, the salmon arm the I superfamily, and the light blue arm the L superfamily. The green arm of the clade of superfamily H indicates that the laccase CB10_180.4889 belongs to the CopA class.

**Figure 3 ijms-24-09785-f003:**
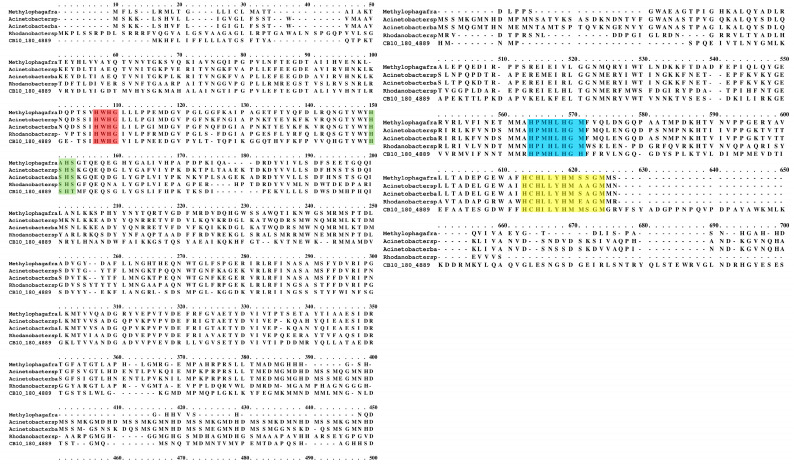
Multiple-sequence alignment and conservation scores of laccases belonging to the CopA family and our Lac_CB10 using the Praline software [31]. The scoring scheme operates from 0 to 10, with 0 being the least conserved position and 10 being the most conserved position, and the asterisks indicate identical residuals between the sequences. The highlighted regions within the boxes indicate the motifs that form the four copper ligands and are highly conserved in laccases: L1 (H-W-H-G, red), L2 (H-S-H, green), L3 (H-P-M-H-L-H-G-H, blue) and L4 (H-C-H-I-L-Y-H-M-M-S-G-M, yellow).

**Figure 4 ijms-24-09785-f004:**
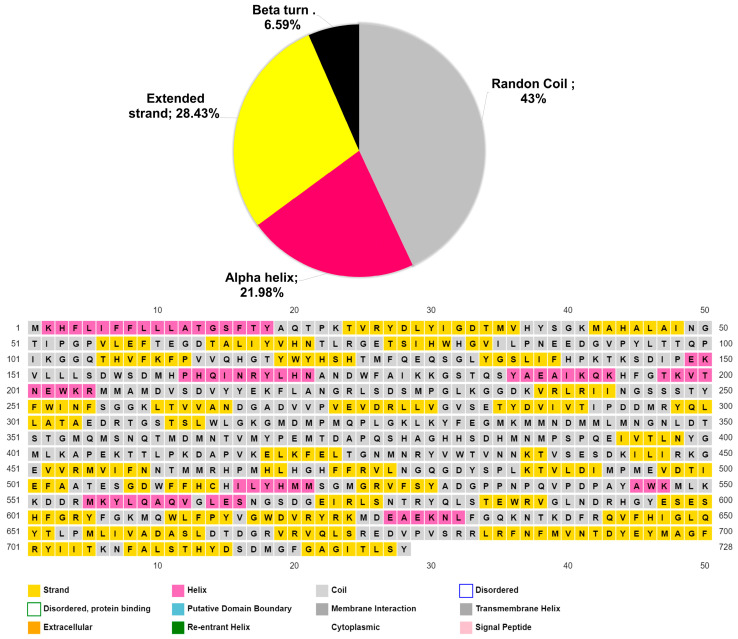
Pie chart representing the percentages of secondary structures present in Lac_CB10, including random coils, colored in gray (43%), beta strands in yellow (28.43%), alpha helices in pink (21.98%) and beta turns in black (6.59%), and a map of the secondary structure of Lac_CB10.

**Figure 5 ijms-24-09785-f005:**
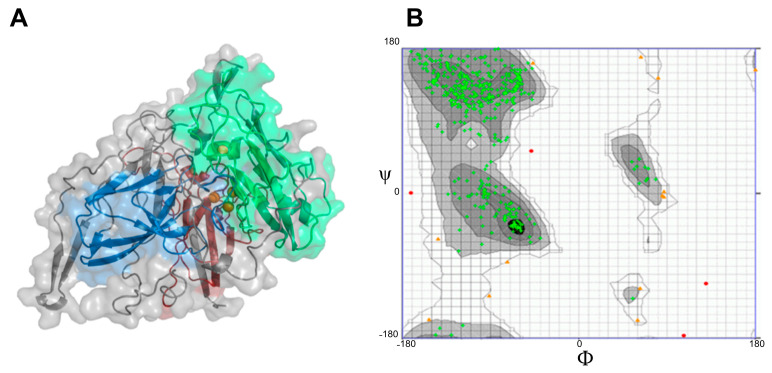
Structural prediction for Lac_CB10. (**A**) Three-dimensional model of Lac_CB10 (template PDB ID 3pps with 30.84% identity) representing the three cupredoxin domains in cartoon and surface forms, with the Cu-oxidase 3 domain colored ruby (dark red), the Cu-oxidase domain colored pale green (green) and the Cu-oxidase 2 colored sky blue (blue). The portions colored in gray represent the amino acids that were not modeled as belonging to the domains, and the four copper atoms are represented as orange spheres. (**B**) Ramachandran plot of the Lac_CB10 structure obtained by homology, where the green dots indicate the amino acids present in favorable regions, representing 96.14% of the entire structure.

**Figure 6 ijms-24-09785-f006:**
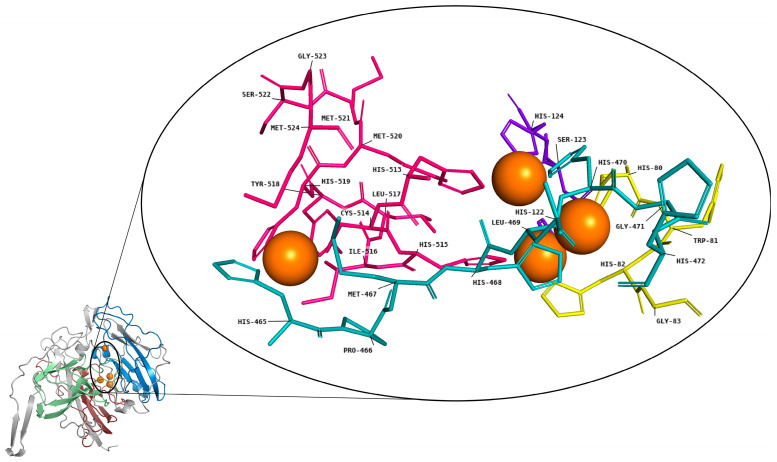
Three-dimensional model of Lac_CB10 (template: PDB ID 3pps with 30.84% identity) indicating the four copper atoms, represented by orange spheres, and the four motifs, represented in stick format, that coordinate these atoms and their respective residues. Motif 1 (H-W-H-G) is colored yellow, motif 2 (H-S-H) is colored violet, motif 3 (H-P-M-H-L-H) is colored cyan and motif 4 (H-C-H-I-L-Y-H-M-M-S-G-M) is colored hot pink.

**Figure 7 ijms-24-09785-f007:**
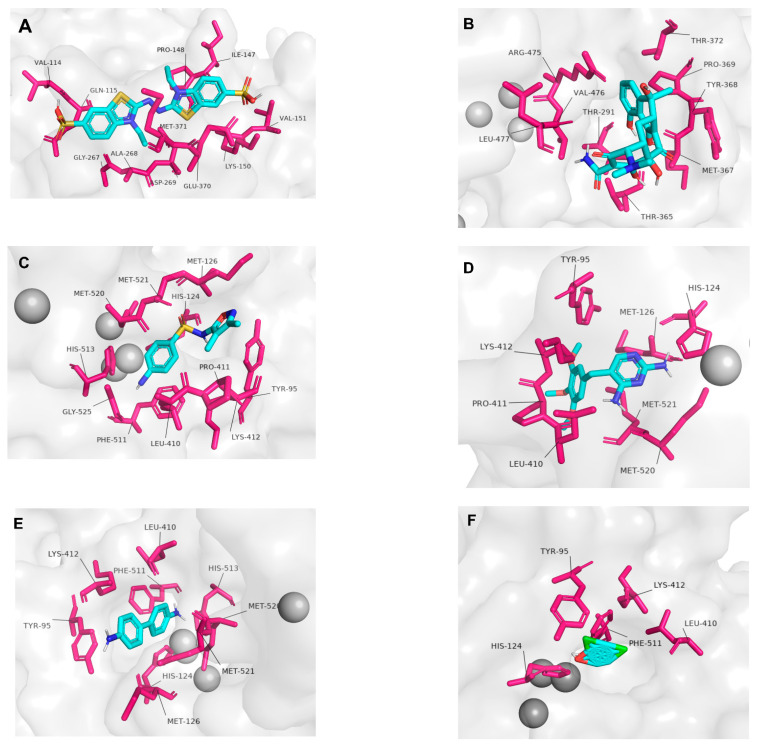
The three-dimensional images (**A**–**F**) show the ligand catalysis inside the pocket. Carbon atoms are represented by blue cyan sticks, oxygen atoms by red sticks, hydrogen atoms by white sticks, nitrogen atoms by dark blue sticks, sulfur atoms by gold sticks, chlorine by green sticks and catalytic residues are shown as pink sticks. The complexes include: (**A**) Laccase–ABTS complex. (**B**) Laccase–tetracycline complex. (**C**) Laccase–sulfisoxazole complex. (**D**) Laccase–trimethoprim complex. (**E**) Laccase–benzidine complex. (**F**) Laccase-2,4–DCP complex.

**Figure 8 ijms-24-09785-f008:**
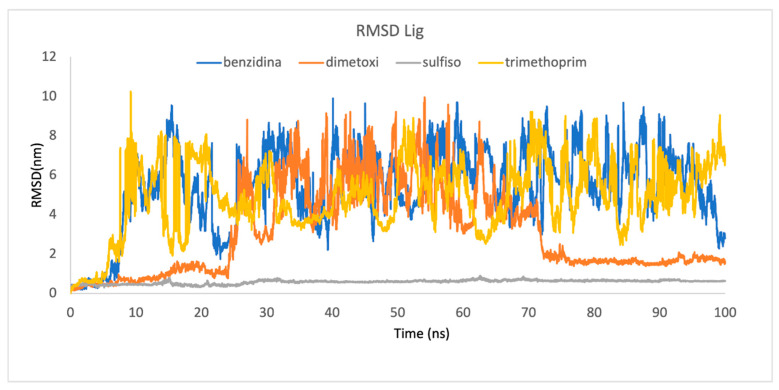
The ligand RMSD as a function of time.

**Figure 9 ijms-24-09785-f009:**
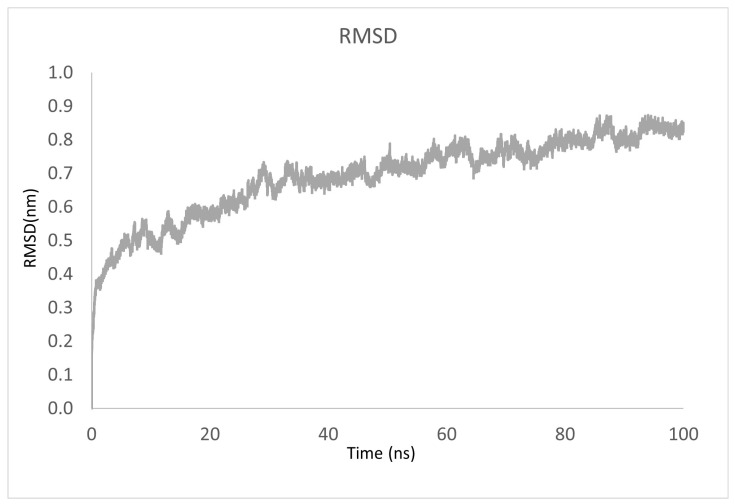
Backbone RMSD as a function of time: sulfiso interacting with Lac-CB10.

**Figure 10 ijms-24-09785-f010:**
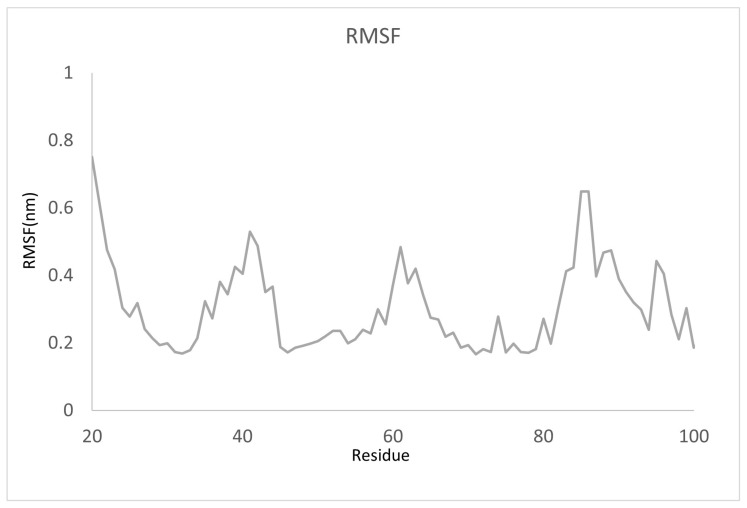
RMSF as functional amino acid residues: sulfiso interacting with Lac-CB10.

**Figure 11 ijms-24-09785-f011:**
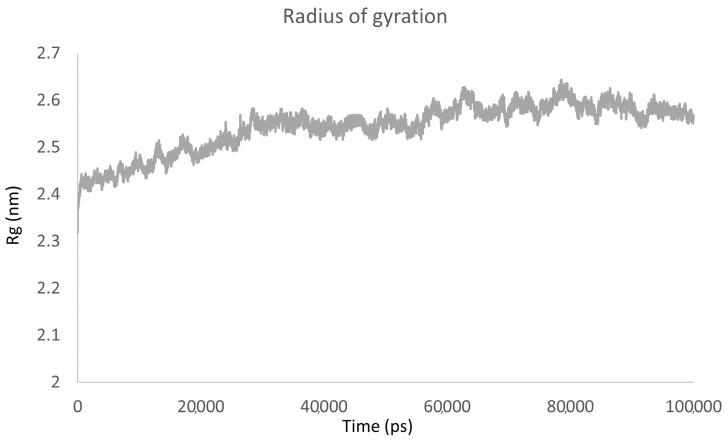
Rg as a function of time.

**Figure 12 ijms-24-09785-f012:**
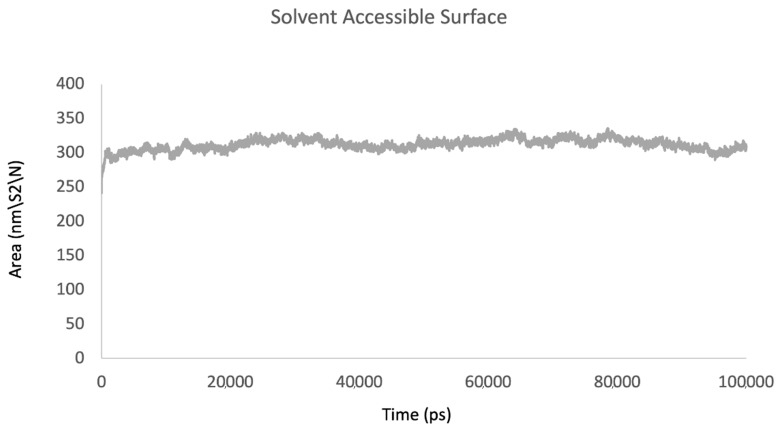
Protein-solvent-accessible surface area as a function of time.

**Table 1 ijms-24-09785-t001:** Gibbs and RMSD free energy values for each laccase–ligand complex evaluated.

Ligand	kcal.mol^−1^
ABTS	−6.9
2,4-Dichlorophenol	−5.9
Benzidine	−6.4
Sulfisoxazole	−6.7
Tetracycline	−8
Trimethoprim	−6.1

**Table 2 ijms-24-09785-t002:** Amino acids involved in ligand interactions in Lac_CB10–ligand complexes. Residues shared by the same ligands are colored with the same color.

Ligand	ABTS	2,4-DCP	Trimethoprim	Tetracyclina	Sulfisoxazole	Benzidine
Amino acid residues	Gln115 Gly267 Met371 Ala268 Asp269 Pro148 Ile147 Val114 Lys150 Glu370 Val151	His124 Phe511 Tyr95 Leu410 Lys412	His124 Met520 Met521 Lys412 Leu410 Pro411 Tyr95 Met126	Leu477 Val476 Thr372 Tyr368 Pro369 Arg475 Thr365 Thr291 Met367	His124 His513 Met521 Gly525 Phe511 Leu410 Met250 Met126 Tyr95 Pro411 Lys412	His124 His513 Met520 Met521 Met521 Phe511 Leu410 Tyr95 Lys412 Met126

## Data Availability

Not applicable.

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
