# Peer review of "Molecular Docking of Lac_CB10: Highlighting the Great Potential for Bioremediation of Recalcitrant Chemical Compounds by One Predicted Bacteroidetes CopA-Laccase"

_ijms, 2023, doi:10.3390/ijms24129785_

Round 1

Reviewer 1 Report (New Reviewer)

The article titled "Molecular docking of Lac_CB10: highlighting the great potential for bioremediation of recalcitrant chemical compounds by one predicted Bacteroidetes CopA-laccase" is an interesting one. It does accomplishes a good degree of communication in a clear fashion. 

However, it is my opinion that it needs some improvement. 

I will try to make suggestions that should be accesible to the authors in the window of time available for publication. I also urge the authors to widen this window in order to put out the best possible work. I don't mind waiting for a revised manuscript when quality is improved. 

On to the notes. 

1) Modelling, this is the weakest link as everything else depends on the quality of the initial model. So my suggestion is, try to generate a secondary model using AlphaFold (https://www.rbvi.ucsf.edu/chimerax/data/alphafold-nov2021/af_sbgrid.html, https://youtu.be/le7NatFo8vI) that supports the original model. My concern is that a ~33% homology is quite low, last time I read on homology models, 60% was the limit for such modeling accuracy. 

2) Docking, an exhaustiveness of 10 seems awfully low for a protein model as big as this laccase (~700 residues). I am sure docking results contain a warning about search space being too large (vide infra).

"WARNING: The search space volume > 27000 Angstrom^3 (See FAQ)"

Increase exhaustiveness to 100 or more. An alternative would be to increase exhaustiveness as well as restrict search space to the neighborhood of the initial docking results. That would make it efficient. 

In this paper (https://doi.org/10.1080/10934529.2015.1038179) observes two different channels found when docking ligands to a laccase. In that work they use an exhaustiveness of 1000. 

3) Ligand preparation (2.3), this section is sparse but  it is missing if there was any reason to select (or not to) specific protonation states of the various nitrogens or sulfate present. This is mentioned for molecular dynamics and no note to indicate they are the same ligand (chargewise) or not. 

Also, the title of this section seems to be just copied from the Methods to the results (3.3) yet the results do not mention anything about the ligand preparation. 

4) Molecular dynamics, I have simulated metalloproteins using AMBER and it is not an easy task. In fact, if the systems is not prepared correctly the Cu atoms do not stay in place. For example, CYS514 is likely to be  deprotonated. No mention that any steps were taken to assure the stability  of the ions in the catalytic site nor that they stay in the same position during the simulation. This is important to assess the quality and accuracy of the results. 

Some colleagues argue that molecular dynamics for a protein-ligand pair should be run and analyzed as triplicates. Others, prefer 5. And, for energetic analysis, sometimes up to 25 replicates. I do recommend 3 replicates at least so the results can be considered representative. 

A set of control simulations (the enzyme without ligand) should also be ran to compare on figures 9, 10, 11 and 12.

5) Molecular dynamics part 2,  figures 8 to 12 only describe the system in a superficial fashions. For example, they do not give any indication of what happens to the protein-ligand interactions.

Are the interaction energies in the same ranking as what was obtained by docking?

Are the ligands stable in the surface or leaving the binding site to find a different one or being solubilized?

Is there an specific interactions (hydrogen bonds, VdW contacts) that are maintained for the 100 ns?

It is my humble opinion that the authors are in position to answer this questions.

For general yet important comments:

Figure 3 must be edited to emphasize what the authors want to show. As it stands now, it emphasizes conservation but makes it difficult to read. 

Figure 1 and 4 maybe be simplified into a single figure. 

Table 1. Kcal.mol-1  should have a lowercase k (kcal.mol-1)

The note about RMSD is meaningless as the first, best result in vina is always RMSD 0.0. As far as I know, no vina can use a user-defined reference ligand to calculate and RMSD. ADFR can do that.

Figure 7. 2,4-DCP should be fired to represent the correct ligand or to show what molecule was actually docked. Not the unrecognizable mess show in F.

Figure 10 was restricted to 100 residues Why not perform the other analyzes also restricted to these 100 residues?

Also, is the numbering correct? As I gather from figure 1 and 3, the domain analyzed is located between residues 36 to 146. That is 110 residues and not, as stated in figure 10, from 20 to 100. 

I do hope the authors find these comments enriching as that is the spirit I have behind them. Good luck.

In general, the use of English language is poor several places. "Amino acids" instead of residues, or "amino acid residues" Residues should suffice. 

As a non-native English speaker, I do recommend a thorough review of the language to make it as clear and easy to read as possible. 

Author Response

I hope this letter finds you well. First and foremost, we would like to express our gratitude for your patience and understanding regarding the initial extension of our submission deadline and, subsequently, for allowing us to make a new manuscript submission in order to perform the suggested molecular dynamics analyses, which we believe have greatly enriched our work.

Considering the time constraints imposed by the doctoral context of the first author, Bárbara B. Buzzo, which requires publication before the imminent deadline, and taking into account the extensive time required to generate a new structural model and perform additional molecular dynamics analysis, unfortunately, we are unable to address, at least in the scope of this study, the suggestions proposed by reviewer 1, despite their evident importance.

After careful consideration and discussions among the authors, we concluded that the most appropriate approach to address these restrictions, while acknowledging the important reservations raised by reviewer 1, is to explicitly mention the technical limitations of this study within the publication itself, as we have highlighted. in the conclusions of our work the need for future investigations that can clarify the remaining questions, either through in silico studies, or through biochemical studies of the enzyme expressed in a heterologous host and crystallographic characterization of the enzyme.

It is important to highlight that we are currently engaged in the biochemical characterization of the laccase under study, which exhibits 100% relative activity at extreme optimal pH and temperature: pH 10.0 and 90 °C, respectively. This characteristic represents a significant advancement in our study, providing a solid foundation for future investigations. After the publication of the in silico analyses and the biochemical characterization, our next step will involve a comprehensive validation study of the results through in vitro assays. We recognize the importance of these subsequent steps for the validation and robustness of our findings.

Although we were unable to implement all of the reviewers' suggestions within the stipulated timeframe, we believe that our current approach will provide valuable insights to the scientific community and pave the way for future studies. All other suggestions have been duly incorporated.

We are available to provide additional information or clarify any questions from the editorial team regarding our work, and we appreciate the time and attention given to our letter and manuscript.

Reviewer 2 Report (New Reviewer)

In this paper, the authors performed an extensive bioinformatics analysis of bacterial laccases with the great potential for bioremediation of recalcitrant chemical compounds. The subject was presented in a rather interesting way. This manuscript requires minor revisions before it is ready for publication. Further detailed comments for consideration are provided below.

Comment 1#

Figure 3 is not clear, please improve the quality of the figure.

Comment 2#

Complex F in the figure 7 is not clear. Correct the structure of the ligand.

Comment 3#

Correct the Figure 9

Comment 4#

Expand the captions under the figures to make them clearer. If you stay only with sulfiso, mark it in the captions and remove it from the figures (9, 10,11,12)

Comment 5#

Correct the references according to the journal's guidelines. Italic for microorganism name.

Comment 6#

In the Table S3  “Organismo” instead of “Organisms”

Comment 7#

In the Figure S1  “cu-oxidase.” instead of “Cu-oxidase.”

Author Response

I hope this letter finds you well. First and foremost, we would like to express our gratitude for your patience and understanding regarding the initial extension of our submission deadline and, subsequently, for allowing us to make a new manuscript submission in order to perform the suggested molecular dynamics analyses, which we believe have greatly enriched our work.

Considering the time constraints imposed by the doctoral context of the first author, Bárbara B. Buzzo, which requires publication before the imminent deadline, and taking into account the extensive time required to generate a new structural model and perform additional molecular dynamics analysis, unfortunately, we are unable to address, at least in the scope of this study, the suggestions proposed by reviewer 1, despite their evident importance.

After careful consideration and discussions among the authors, we concluded that the most appropriate approach to address these restrictions, while acknowledging the important reservations raised by reviewer 1, is to explicitly mention the technical limitations of this study within the publication itself, as we have highlighted. in the conclusions of our work the need for future investigations that can clarify the remaining questions, either through in silico studies, or through biochemical studies of the enzyme expressed in a heterologous host and crystallographic characterization of the enzyme.

It is important to highlight that we are currently engaged in the biochemical characterization of the laccase under study, which exhibits 100% relative activity at extreme optimal pH and temperature: pH 10.0 and 90 °C, respectively. This characteristic represents a significant advancement in our study, providing a solid foundation for future investigations. After the publication of the in silico analyses and the biochemical characterization, our next step will involve a comprehensive validation study of the results through in vitro assays. We recognize the importance of these subsequent steps for the validation and robustness of our findings.

Although we were unable to implement all of the reviewers' suggestions within the stipulated timeframe, we believe that our current approach will provide valuable insights to the scientific community and pave the way for future studies. All other suggestions have been duly incorporated.

We are available to provide additional information or clarify any questions from the editorial team regarding our work, and we appreciate the time and attention given to our letter and manuscript.

Reviewer 2,

We thank the reviewer for the valuable suggestions provided. All the points raised have been addressed and incorporated into the text, with all modifications being tracked using the "Track Changes" function in MS Word, if applicable. We sincerely appreciate your time and contribution, which have been crucial in improving the quality of our work.

Reviewer 3 Report (New Reviewer)

This study is interesting, well developed,  coherent and consistent. However it presents some issues that require Authors' intervention. On the whole, the Authors use a correct English language, although many sentences are decidedly too long and should be changed into shorter sentences. I have noted just a few examples in the list here below. Moreover, the References do not meet the Journal rules for bibliography, provided that many entries contain unnecessary dates of publication, written in Portuguese. 

Lines 24-26 Please check the sentence syntax.

L 90-93 Please check this sentence.

L 294 Do you mean color disappearance (bleaching) or color change?

L 416-417 Why have the Authors specified in parentheses additional color definitions?

L 422-428 This sentence is correct but is decidedly too long. I suggest expressing the same concepts but in two or three sentences instead.

L 428-433 The same as above

L 475-479 Please reconsider the legend of Figure 7, where no red and blue spheres representing oxygen and hydrogen atoms can be seen. Moreover, I was somewhat astonished in inspecting Figure 7-F (is that 2,4-dichlorophenol?!)

L 507 “… environmental effects on the environment…”

L 511 What is the similarity between benzidine and 2,4-DCP?

The English language is fine with concern to vocabulary and syntax. However, many sentences are decidedly too long and should be reconsidered and changed in more shorter sentences.

Author Response

I hope this letter finds you well. First and foremost, we would like to express our gratitude for your patience and understanding regarding the initial extension of our submission deadline and, subsequently, for allowing us to make a new manuscript submission in order to perform the suggested molecular dynamics analyses, which we believe have greatly enriched our work.

Considering the time constraints imposed by the doctoral context of the first author, Bárbara B. Buzzo, which requires publication before the imminent deadline, and taking into account the extensive time required to generate a new structural model and perform additional molecular dynamics analysis, unfortunately, we are unable to address, at least in the scope of this study, the suggestions proposed by reviewer 1, despite their evident importance.

After careful consideration and discussions among the authors, we concluded that the most appropriate approach to address these restrictions, while acknowledging the important reservations raised by reviewer 1, is to explicitly mention the technical limitations of this study within the publication itself, as we have highlighted. in the conclusions of our work the need for future investigations that can clarify the remaining questions, either through in silico studies, or through biochemical studies of the enzyme expressed in a heterologous host and crystallographic characterization of the enzyme.

It is important to highlight that we are currently engaged in the biochemical characterization of the laccase under study, which exhibits 100% relative activity at extreme optimal pH and temperature: pH 10.0 and 90 °C, respectively. This characteristic represents a significant advancement in our study, providing a solid foundation for future investigations. After the publication of the in silico analyses and the biochemical characterization, our next step will involve a comprehensive validation study of the results through in vitro assays. We recognize the importance of these subsequent steps for the validation and robustness of our findings.

Although we were unable to implement all of the reviewers' suggestions within the stipulated timeframe, we believe that our current approach will provide valuable insights to the scientific community and pave the way for future studies. All other suggestions have been duly incorporated.

We are available to provide additional information or clarify any questions from the editorial team regarding our work, and we appreciate the time and attention given to our letter and manuscript.

Reviewer 3,

We thank the reviewer for the valuable suggestions provided. All the points raised have been addressed and incorporated into the text, with all modifications being tracked using the "Track Changes" function in MS Word, if applicable. We sincerely appreciate your time and contribution, which have been crucial in improving the quality of our work.

Question 3,

L 294:

Do you mean color disappearance (bleaching) or color change?

Answer:

Thank you for pointing out this ambiguous wording in the sentence. We mean "disappearance" of color, as the laccase is capable of degrading the mentioned dyes.

Question 7,

L 475-479:

Please reconsider the legend of Figure 7, where no red and blue spheres representing oxygen and hydrogen atoms can be seen. Moreover, I was somewhat astonished in inspecting Figure 7-F (is that 2,4-dichlorophenol?!).

Answer:

The structure presented in Figure 7 was downloaded from the PubChem database and represents 2,4-dichlorophenol. Its position has been modified in the manuscript for better visualization.

Question 9,

L 511:

What is the similarity between benzidine and 2,4-DCP?

Answer:

Similar to benzidine, 2,4-dichlorophenol (2,4-DCP) is a synthetic phenolic compound widely used in pesticides. It is also found in water bodies due to unintentional discharge and leaching, which is associated with the compound's persistence. Both benzidine and 2,4-dichlorophenol are chemical compounds employed in various industrial applications. They are recognized as toxic and can potentially pose risks to human health and the environment if not handled or disposed of properly.

This manuscript is a resubmission of an earlier submission. The following is a list of the peer review reports and author responses from that submission.

Round 1

Reviewer 1 Report

Comments on ijms-1878233:

The current manuscript reports a computational investigation of the protein-ligand binding case involving the copper-binding putative laccase CB10_180.4889 (Lac_CB10 in the title). The tertiary structure of Lac_CB10 is obtained from homology modelling due to the absence of experimentally deposited structures of related species. Molecular docking with Autodock is performed to screen ligands and the affinity rank of tetracycline > ABTS > sulfisoxazole > benzidine > trimethoprim > 2,4-DCP is reached. The detailed interaction between protein and ligands is also investigated and important residues contributing significantly to the stabilization effect are summarized. Although the aim of the current work is of scientific importance and seems interesting to the field, the design of the research requires significant improvements. The text is also written without caution and careful proofreading should be finished before submitting the revised manuscript.

Some abbreviations (e.g., DHHS on page 2) are defined but never mentioned later. In that case, these abbreviations should not be defined at all. On the other hand, some abbreviations (e.g., LBMP) are used without definition. Further, many abbreviations are not defined properly. For example, in the caption of Figure 1, AA is defined as the abbreviation of amino acid number, which seems very strange. The authors are expected to check this abbreviation issue thoroughly.

The authors claim that Leu and Val have aliphatic side chains, which aids in molecular stability. However, I wonder how this statement is reached. They are not widely acknowledged facts in current biological research.

Similarly, the authors state that Asp residues are often found on the protein surface and exposed to water, which is neither widely accepted. There are many Asp residues inside the protein interior. They can form hydrogen bonds (salt bridges) with nearby residues and also modulate the pH-dependent behavior of functional regions.

There are many titratable sites in ligands. For example, there are two -SO3- groups in ABTS. Have the authors computed their pKa values with some tools for drugs? Is it possible that some of them become protonated? Would their protonation alter the docking result?

Similarly, the nitrogen atoms in some ligands could also have protonation-deprotonation equilibria. For example, the -NMe2 group in Tetracycline could be protonated to produce a quaternary ammonium cation center, which could have a significant impact on protein-ligand binding. What is the pKa of the -NMe2 group in this molecule? Could it be protonated under the physiological condition?

The quality of the figures should be improved. Specifically, the resolution of Figure 1b, the font size of Figure 2, both the resolution and font size of Figure 3, the resolution of Figure 4, and the resolution of Figure 7 should be improved. As for the Ramachandran plot in Figure 4b, the axis should be marked as phi/psi.

Many figures are of little scientific importance and should be moved to the supporting material. For example, Figure 8-13 are obtained from simple interaction-map analysis and reporting similar information. These data are not really crucial and should be moved to supporting information (SI). The chemical structures of ligands in Figure 7 are also unimportant and should be moved to SI.

In Table 1, the statement of ‘Gibbs and RMSD free energy values’ does not seem reasonable.

The ‘2. Results and discussions’ and ‘3. Materials and Methods’ sections should be re-ordered. The computational details should be introduced first, after which the computational results are discussed.

The bound structure with the highest affinity is extracted from docking results and is then used for the analysis of protein-ligand interaction. However, the docking scores of the top-n binding modes given by molecular docking (e.g., Autodock Vina and Glide) could be extremely similar. For example, the top-2 or top-3 docking scores could differ by ~0.1 kcal/mol, which indicates that the semi-empirical scoring function used in molecular docking already has trouble differentiating the relative stability of these binding poses. However, the structural RMSD between them could be very huge (e.g., ~2 Angstrom), suggesting significant differences in their structural features. Thus, further molecular simulations are often initiated from the top-n binding poses obtained from docking to validate the stability of these binding modes. After that procedure, the protein-ligand binding pose is expected to be really reliable and can be used for interaction analysis, although it is still possible that the bound structure is stuck in some local minima and fails to explore the true optimal basin. Further rescoring of the docking rank is often performed to re-rank the binding poses to reach a more reliable conclusion of the affinity rank.

Reviewer 2 Report

In their manuscript entitled, “Molecular docking for Lac_CB10: highlights of great potential for bioremediation of recalcitrant chemical compounds for one Bacteroidetes predicted CopA-laccase’’ Buzzo et al. reported an in-silico investigation to understand the general parameters of the enzyme, analyze its conserved regions and tertiary structure, and assess its interaction with compounds of environmental interest by the molecular docking technique. The following concerns should be addressed prior to reconsideration of the paper.

-Line 307: Please consider using "PDB ID" before any PDB code with appropriate referencing.

-Lines 340-342: With the limited similarity value of the Lac_CB10 structure model with the template, it is recommended that the authors perform an MD simulation of the homology model to allow refining the model structure that brings more reliability to the use of the model.

-Line 360: Please consider substituting “coupled” with “docked”.

-Line 367: In addition to ABTS as a positive control, authors should include a non-binding compound as a negative control as well.

-Figure 8B, 9B, 10B, 11B, 12B, and 13B presentation could be improved by adding labels for the residues interacting with ligands and adjusted to be clearly visible in the figure.

-Line 514-516: Authors should provide other analysis such as MD simulation to investigate complexes' stability and to further verify if the ligands stay bound in their binding pocket not drifting away into the solvent.